# Randall's Threadfin Bream (*Nemipterus randalli,* Russell 1986) Poses a Potential Threat to the Northeastern Mediterranean Sea Food Web

Yagmur Akgun [1] and Ekin Akoglu [2,*]

1    Department of Biological Sciences, Middle East Technical University, 06800 Ankara, Türkiye;
     akgun.yagmur@metu.edu.tr
2    Institute of Marine Sciences, Middle East Technical University, 33731 Erdemli, Türkiye
*    Correspondence: eakoglu@metu.edu.tr

**Abstract:** The eastern Mediterranean Sea is one of the most invaded marine ecosystems due to the introduction of Lessepsian species, which migrated from the Red Sea to the Mediterranean Sea following the construction of the Suez Canal. Some of these species may initially appear to be beneficial for fisheries by providing additional income sources for fishers; however, this usually occurs at the expense of negatively impacted native species and, thus, the ecosystem, which leads to greater economic losses for the fisheries in the long term. Therefore, this study aims to quantify the impact of *N. randalli*, which is one of the Lessepsian species with increasing commercial importance for the fisheries, on the food web dynamics in a coastal ecosystem in the northeastern Mediterranean Sea using a mass-balance food web modelling approach by capitalising on field data obtained from trawl samplings conducted within the scope of the study. Results showed that the ecosystem was in a developmental stage and experienced an autotrophic succession. The keystone fish group with a structuring role in the food web was sea breams and porgies. Although *N. randalli* had positive impacts on certain commercially exploited indigenous demersal fish species by mitigating the negative impact of another Lessepsian species, i.e., *Saurida undosquamis* (Richardson, 1848), in the food web, it had a negative impact on the keystone group of the food web, i.e., sea breams and porgies. Therefore, *N. randalli* poses a potential threat to the ecosystem's structure, and the interactions of *N. randalli* with other species in the food web may instigate an ecosystem reorganisation in the future. We suggest targeted fisheries exploitation and incentives for the fishery of *N. randalli* as management strategies to mitigate its negative impacts. However, the mitigating role of *N. randalli* in regulating the negative impacts of *S. undosquamis* could be adversely affected by its increasing exploitation; therefore, future modelling studies should consider scenario simulations to test such effects.

**Keywords:** alien fish species; Ecopath; food web modelling; ecological impact; Randall's threadfin bream

**Key Contribution:** Randall's threadfin bream has a mitigating role against negative impacts of other Lessepsian fish species, whereas it could be a potential instigator of drastic reorganisations in the northeastern Mediterranean Sea food web.



## 1. Introduction

Species that move or are introduced beyond their past or present distribution and are capable of surviving and reproducing in their new environment are called alien species, and alien species that threaten the biological diversity in their new environment are called invasive alien species [1]. The Levantine Sea is considered one of the marine regions most impacted by biological invasions, with an alien-to-native species richness ratio of 0.69 [2], and was tremendously affected by two anthropogenic stressors other than fisheries: the constructions of the Suez Canal and the Aswan Dam. Both have a crucial role in the species migration from the Red Sea to the Mediterranean Sea, which is known as the Lessepsian

migration (named after the engineer and developer of the canal Ferdinand de Lesseps) or Erythraean invasion. The Suez Canal was completed in 1869 to provide a shorter maritime route between the western Pacific, the Indian Ocean and the Mediterranean Sea, and the Atlantic Ocean. Initially, the Suez Canal was 8 m deep, and its depth gradually increased to 24 m. This further deepening of the Suez Canal facilitated the migration of species to the Mediterranean Sea from the Red Sea [3]. Furthermore, the construction of the Aswan Dam in 1965 reduced the fresh water inflow from the River Nile, which is in proximity to the Suez Canal, that previously provided a biogeographic barrier against the Lessepsian migrants [4,5].

The abundance and biomass of invasive fish species have doubled during the last two decades in the Levantine Basin [6]. Increasing sea surface temperatures in the Levantine Sea due to global warming is hypothesised to have facilitated the successful establishment of the invasive species from the Red Sea [7]. Furthermore, the Red Sea has similar conditions to those in the Levantine Sea, i.e., poor nutrient levels and high salinity (38.7 practical salinity unit (PSU) in the Levantine Sea, 40–41 PSU in the northern Red Sea and around 41 PSU in the Gulf of Suez); therefore, the Mediterranean Sea provides a familiar environment for the Lessepsian migrants and, thus, increases their chances of successful establishment [8].

Ecopath with Ecosim (EwE) is one of the most widely adopted food web models that is used to delineate the structure and function of marine food webs and ecosystems [9]. Modelling studies using EwE were previously conducted to represent the food web interactions, impact of fisheries and introduction of alien species in the eastern Mediterranean Sea. In the Aegean Sea, fisheries exploitation was high and the microbial food web influenced the functioning of the ecosystem [10], and different fishery management scenarios indicated an inevitable reduction in pelagic species biomasses [11]. In the Thermaikos Gulf in the north Aegean Sea, changing environmental factors and fishing activities instigated biomass declines in fish assemblages, and the best mitigation option was to decrease the exploitation levels by fisheries [12]. Similarly, the declines in the biomasses and catches of marine living resources were due to the changing environmental factors and fisheries exploitation in the Pagasitikos Gulf located in the central Aegean Sea, and a reduction in fisheries exploitation levels was suggested to mitigate adverse changes in the ecosystem [13]. In the Saronikos Gulf in the central Aegean Sea, fisheries exploitation levels were unsustainable and exerted negative impacts on a wide spectrum of species [14]. In a modelling study of the whole Aegean Sea, similar to previous efforts, high fisheries exploitation and its impact on the ecosystem was prominent [15]. Along the coasts of western and southwestern Cyprus island, which is located in the northeastern Mediterranean Sea, alien species had significant impacts on phytobenthos, and eels and morays in the ecosystem [16]. In a study that compared the ecosystem conditions on the Israeli coasts between the 1990s and 2010s, the increasing impact of alien species were evident as the contribution of alien species to the fish biomass and catch in the region increased [17]. In the Mersin Bay in the Cilician Basin, alien species played a key role in benthic–pelagic coupling in the food web, and fisheries mediated the role of alien species [18]. Overall, the impact of fisheries was prevalent in previous modelling studies on the eastern Mediterranean Sea and had the potential to mediate the roles of fish species in the ecosystem. Previously, fisheries, i.g., strategic overfishing, were suggested as a management tool to mitigate the adverse effects of alien and/or invasive species in the food web, e.g., lionfish (Pterois miles, Bennett, 1828) in the northwest Atlantic and invasive freshwater crayfish in North America [19]. Therefore, fisheries can play a role in regulating the negative impacts of alien species in the northeastern Mediterranean Sea.

One of the common Lessepsian species observed on the Turkish coasts is *N. randalli*. It has recently become abundant in the catch composition [20] and is a commercially important fish species in Turkey. Following its first recorded sighting in Haifa Bay in 2005 [21], it was observed on the Lebanon coast [22], in İskenderun Bay [23], Gökova Bay [24] and İzmir Bay [25] in 2007, 2007, 2011 and 2016, respectively. *N. randalli* was considered a species with a high potentiality for being invasive in the Mediterranean

Sea [26,27]. Furthermore, *N. randalli* is famous for its resemblance to one of the iconic commercial species, i.e., *P. erythrinus*, in the region, and has increasingly been marketed as *P. erythrinus* [28]. Hence, it is critical to assess the impact of *N. randalli* on the food web of the Levantine Sea and explore possible mitigation strategies, considering its interactions with commercially important indigenous species.

In this study, we investigated the impacts of *N. randalli* on the food web and native species in the Lamas (Limonlu) region in the Cilician Basin. We further identified the vulnerable native species that could be negatively impacted by further establishment of *N. randalli* and proposed possible mitigation strategies. Specifically, we sought answers to three fundamental questions related to the role of *N. randalli* in the study region: (i) what is the impact of *N. randalli* on the indigenous species in the Lamas region? (ii) how can *N. randalli* affect the food web dynamics? and (iii) how could the impacts of *N. randalli* be mitigated?

## 2. Materials and Methods

### 2.1. Study Area

The Lamas (Limonlu) marine region is located towards the west of Gulf of Mersin, an important coastal area where the wide continental shelf in the eastern part of the Cilician Basin starts to narrow towards the west to the Pamphylia and Lycia basins located in the vicinities of Anamur and Antalya (Figure 1). Therefore, the coastal area of the Lamas region is an amalgamation of the characteristics of narrow- and wide-shelf marine coastal regions and a habitat to a diverse range of fish assemblages that is typical to both coastal regions. The Lamas River also discharges in close proximity to the west of the study area, and therefore provides a productive marine environment due to the nutrients provided via its flow. These characteristics of the region make the coastal zone of the Lamas region a perfect area for investigating the impacts of alien species in a typical northeastern Mediterranean Sea marine coastal ecosystem.

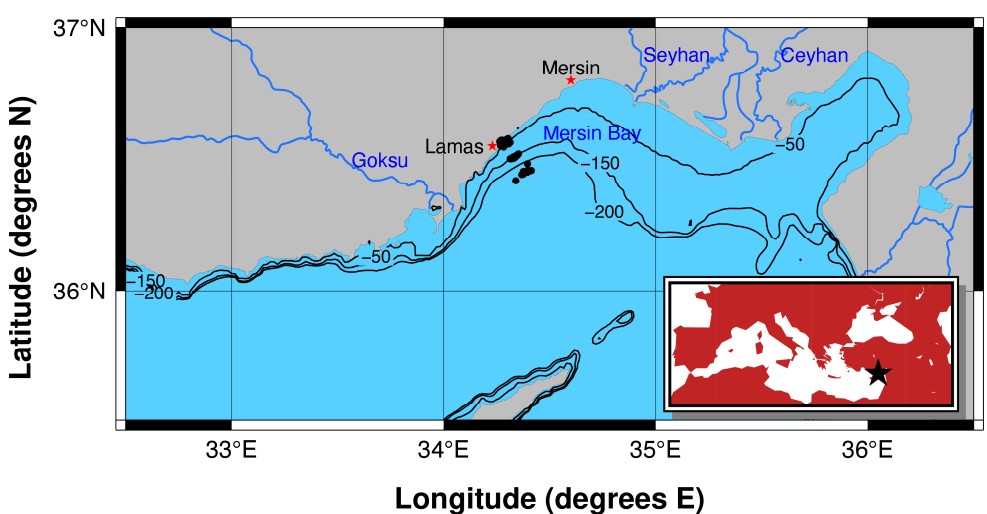

**Figure 1.** The location of the study in the Mediterranean Sea (denoted with a star in the map, located lower right) and sampling locations (black dots) in the Lamas marine region (larger map).

### 2.2. Sampling

The study site covered 1.76 km$^2$ of the coastal region in front of the town of Limonlu in the Lamas region of Erdemli, Mersin. Fish samples were collected monthly from 68 stations between January 2019 and January 2020 (Figure 1). Here, 18 mm trawl nets were used at depths extending from 16 m to 210 m, classified under four depth strata at 0–49 m (26 hauls), 50–99 m (15 hauls), 100–149 m (13 hauls) and 200–249 m (14 hauls). For each trawl haul, the sampling time was 15 min for the 0–49 m depth stratum, 30 min for 50–99 and 100–149 m depth strata, and 60 min for the 200–249 m depth stratum. The

samples were collected with R/V Lamas of the Institute of Marine Sciences, Middle East Technical University.

*2.3. Stomach Content Analysis*

The stomach contents of *N. randalli* specimens were studied using individuals collected from trawl hauls to parameterise the diet composition of the species for the modelling study. First, the lengths of the specimens were measured to the nearest millimetre. The upper tail of the caudal fin of *N. randalli* is elongated, like a filament; therefore, fork lengths were measured as suggested in the literature [29]. Then, the specimens were gutted and the stomachs were extracted, weighed and stored in the freezer for later analysis. At least three stomach samples from each length group were sampled from a total of 16 length classes, ranging between 6 cm and 21 cm, to avoid bias for certain length classes. A total of 64 stomach samples were analysed: 22 from spring, 16 from summer, 11 from autumn and 15 from winter. Prior to identification, the wet weights of stomachs were measured by a precision scale (Precisa XB 220A) with a sensitivity of 0.01 mg. Afterwards, the stomach membranes were removed from the stomach contents. The stomach contents were rinsed with water to remove microscopic organisms and placed on blotter paper to remove excess water before weighing. For the identification of samples, a light microscope (Olympus SZX12) was used at a $20\times$ magnification. Finally, each item in the stomach contents was weighed separately and relative diet compositions by weight for each identified stomach sample were calculated. Contents such as endoparasites, unidentified digested organic material and lophotrochozoans were grouped under the detritus group.

*2.4. Modelling Approach*

Ecopath with Ecosim (EwE) version 6.6.8 ([9], available at www.ecopath.org (accessed on 16 March 2023) was used to set up a food web model of the study area. The Lamas region Ecopath model used in this study is an updated version of the model by [30]. Ecopath is the mass-balance trophodynamic model of the EwE modelling suite and is based on two master equations that ensure mass and energy balance. The first master equation ensures mass balance as

$$P_i - M2_i - M0_i - E_i - Y_i - BA_i = 0$$

where $P_i$ is the total production of functional group or species $i$, $M2_i$ is the predation mortality rate of $i$, $M0_i$ is the other mortality rate of $i$ due to diseases, starvation or old age, $E_i$ is the net migration rate of $i$, $Y_i$ is the total fishery catch rate of $i$, and $BA_i$ is the biomass accumulation rate of $i$.

This equation can be re-expressed as

$$B_i * \left(\frac{P}{B}\right)_i - \sum_{j=1}^{n} B_j * \left(\frac{Q}{B}\right)_j * DC_{ji} - (1 - EE_i) * B_i * \left(\frac{P}{B}\right)_i - E_i - Y_i - BA_i = 0$$

where $B_i$ is the biomass of functional group or species $i$, $(P/B)_i$ is the production-to-biomass ratio of $i$, $(Q/B)_i$ is the consumption-to-biomass ratio of $i$, $DC_{ji}$ is the fraction of prey, $i$, in the diet of predator $j$, and $EE_i$ is the ecotrophic efficiency of $i$, that is, the fraction of the production of $i$ that is not exported and is used in the system. In addition to the specifications of the relative diet composition matrix (DC), Ecopath requires three of the four parameters, namely $B$, $P/B$, $Q/B$ and $EE$, to be specified. Furthermore, catches for the exploited species/groups can be specified.

Ecopath ensures the energy balance of a functional group or species as

$$Q_i = P_i + R_i + E_i$$

where $Q_i$, $P_i$, $R_i$ and $E_i$ are the consumption, production, respiration and egestion of group $i$, respectively.

Fourteen functional groups and six species, as well as a detritus compartment, were defined in the model (Table 1). The species/groups included in the model met three criteria: (i) having a direct prey–predator interaction with *N. randalli*, (ii) having an indirect relationship, i.e., trophic competition, with *N. randalli*, or (iii) being first-order prey or predators of groups/species that interacted directly or indirectly with *N. randalli*. Functional groups in the model were constituted based on similarity of their diets and predators.

**Table 1.** Species and functional groups in the Lamas region Ecopath model.

| Functional Group | Species and Taxa Included |
|---|---|
| Detritus | Sediment and water-column detritus |
| Phytoplankton | Planktonic algae |
| Zooplankton | Fodder micro- and mesozooplankton |
| *Nemipterus randalli* | *N. randalli* |
| Other benthic invertebrates | *Philine* spp., *Anseropoda placenta* (Pennant, 1777); *Echinaster (Echinaster) sepositus* (Retzius, 1783); *Pennatula phosphorea* Linnaeus, 1758; *Pennatula rubra* (Ellis, 1764); *Antedon* spp.; *Coscinasterias tenuispina* (Lamarck, 1816) Gastropoda Bivalvia |
| Polychaetes | All taxa |
| Crabs | *Pagurus prideaux* Leach, 1815; *Medorippe lanata* (Linnaeus, 1767); *Charybdis (Archias) longicollis* Leene, 1938 |
| Shrimps and prawns | *Penaeus japonicus* Spence Bate, 1888; *Penaeus kerathurus* (Forskål, 1775); *Parapenaeus longirostris* (Lucas, 1846); *Squilla mantis* (Linnaeus, 1758); *Erugosquilla massavensis* (Kossmann, 1880) |
| Octopuses, cuttlefish and squids | *Eledone moschata* (Lamarck, 1798); *Octopus vulgaris* Cuvier, 1797; *Sepia officinalis* Linnaeus, 1758; *Illex coindetii* (Vérany, 1839); *Loligo vulgaris* Lamarck, 1798; *Rhombosepion elegans* (Blainville, 1827); *Rhombosepion orbignyanum* (Férussac, 1826); *Sepietta oweniana* (d'Orbigny, 1841) |
| *Pagellus erythrinus* (Linnaeus, 1758) | *P. erythrinus* |
| *Pagellus acarne* (Risso, 1827) | *P. acarne* |
| Red mullets | *Mullus barbatus* Linnaeus, 1758 and *Mullus surmuletus* Linnaeus, 1758 |
| *Merluccius merluccius* (Linnaeus, 1758) | *M. merluccius* |
| *Gobius* spp. | *Gobius bucchichi* Steindachner, 1870; *Gobius niger* Linnaeus, 1758; *Vanderhorstia mertensi* Klausewitz, 1974 |
| *Saurida undosquamis* (Richardson, 1848) | *S. undosquamis* |
| Sea breams and porgies | *Boops boops* (Linnaeus, 1758); *Dentex macrophthalmus* (Bloch, 1971); *Diplodus annularis* (Linnaeus, 1758); *Diplodus sargus* (Linnaeus, 1758); *Diplodus vulgaris* (Geoffroy Saint-Hilaire, 1817); *Lithognathus mormyrus* (Linnaeus, 1758); *Evynnis ehrenbergii* (Valenciennes, 1830); *Pagrus pagrus* (Linnaeus, 1758); *Sparus aurata* Linnaeus, 1758; *Spicara flexuosum* Rafinesque, 1810; *Spicara smaris* (Linnaeus, 1758) |
| *Serranus* spp. | *Serranus hepatus* (Linnaeus, 1758); *Serranus cabrilla* (Linnaeus, 1758) |
| Leiognathidae | *Equulites elongatus* (Günther, 1874); *Equulites klunzingeri* (Steindachner, 1898) |
| Clupeidae | *Dussumieria elopsoides* Bleeker, 1849; *Sardina pilchardus* (Walbaum, 1792); *Sardinella aurita* Valenciennes, 1847; *Sardinella maderensis* (Lowe, 1838) |
| *Engraulis encrasicolus* (Linnaeus, 1758) | *E. encrasicolus* |
| Horse mackerels | *Trachurus mediterraneus* (Steindachner, 1868) and *Trachurus trachurus* (Linnaeus, 1758) |

The initial conditions, i.e., the biomasses of species and functional groups except phytoplankton, zooplankton, polychaetes and detritus, of the Ecopath model were calculated from trawl sampling conducted in the study, and the rest were obtained by capitalising

on published literature in the region and, if necessary, data from adjacent areas (Table A1). Biomasses were calculated using the swept area method using data from the monthly trawl surveys. The swept area (a) was estimated by

$$a = D * h * X_2$$

where $D$ is the distance covered during each trawl tow, $h$ is the length of the head rope and $X_2$ is the fraction of the head rope length that is equal to the width of the path swept by the trawl net. The distance covered was calculated as

$$D = 60 * \sqrt{(Lat_1 - Lat_2)^2 + (Lon_1 - Lon_2)^2 * \cos^2(0.5 * Lat_1 + Lat_2)}$$

where $Lat_1$ and $Lon_1$ are the starting latitude and longitude, and $Lat_2$ and $Lon_2$ are the final latitude and longitude of the trawl operation, respectively.

The catch per unit of area (CPUA) of species in the haul was calculated as

$$CPUA = \frac{C_W}{a}$$

where $C_w$ is the catch weight and $a$ is the swept area by the trawl. Fishing gear cannot retain all the fish in the environment; therefore, assuming that there is a relationship between the CPUA and the true biomass of the fish, CPUA values should be converted to biomass values using a proportionality constant [31]. The CPUA values were converted to biomasses as

$$B = \frac{CPUA}{X_1}$$

where $X_1$ is the proportion of the fish in the path of the tow that was retained by the fishing gear. For practicality, we assumed that all fish in the path of the tow were retained [32].

The *P/B* is assumed to be equal to the total mortality (Z) under steady-state conditions [33]. Therefore, if no literature data were available from the study area, we calculated *P/B* ratios for teleost fishes as

$$\ln Z = 1.46 - 1.01 * \ln(A_{max})$$

where $A_{max}$ is the maximum age for the species [34]. The *P/B* ratios of fish functional groups were calculated by averaging the calculated *P/B* ratios of each species in the functional group by its corresponding biomass in the group. Mortalities of other fish groups were obtained from the literature (Table A1). *P/B* ratios of other benthic invertebrates, polychaetes, crabs, shrimps and prawns, and octopuses, cuttlefish and squids groups were obtained from previous studies.

The *Q/B* ratios for fish groups and species were calculated empirically as

$$\log\left(\frac{Q}{B}\right) = 7.964 - 0.204 * \log W_\infty - 1.965 * T' + 0.083 * A + 0.532 * h + 0.398 * d$$

where $W_\infty$ is the asymptotic weight of the fish, $A$ is the aspect ratio of the caudal fin, $T'$ is the mean ambient water temperature of the fish's habitat expressed in 1000/degrees Kelvin, and $h$ and $d$ are the diet parameters depending on the feeding type. If the fish is a carnivore, then $h$ and $d$ are equal to 0; if the fish is a herbivore, the values of $h$ and $d$ are equal to 1 and 0, respectively; and, if the fish is a detritivore, the values of $h$ and $d$ are equal to 0 and 1, respectively [35]. The *Q/B* ratios of functional groups other than fish were taken from previous modelling and empirical studies in the Mediterranean Sea (Table A1). Aspect ratios of all fish groups except *N. randalli* were obtained from the literature [36]. The aspect

ratio of *N. randalli* was calculated by capitalizing on the measurements on the sampled specimens as

$$A = \frac{h^2}{s}$$

where *h* and *s* are the height and surface area of the caudal fin, respectively. Fifty tail samples from individuals that ranged between 4 and 21 cm fork lengths were processed. The tails were photographed with a microscope camera (Olympus DP26). The heights and surface areas of the caudal fins were measured using ImageJ image processing software (https://imagej.nih.gov/ij/, accessed on 7 September 2020).

The relative diet composition of *N. randalli* was calculated within the scope of this study and complemented by available literature in the region, and diet information for other groups and species was obtained from the literature (Table A1). The relative diet composition matrix for the Ecopath model is given in Table A2. All the data sources used to parameterise the Lamas region Ecopath model are listed in Table A1.

Statistical catch data were obtained from the official national landing statistics [37], except Clupeidae, which was obtained from the Sea Around Us project [38]. The statistical data covered all of the Mediterranean coast of Turkey, and did not differentiate geographical regions. Therefore, the annual statistical landings were divided by the total area of the Turkish Exclusive Economic Zone (EEZ) to obtain annual catch rates in tonnes per square kilometre per year. A total area of 72,195 km$^2$ was used to represent the EEZ of Türkiye in the Mediterranean Sea where fisheries operated [38]. Turkish landing statistics did not have records for *N. randalli*; therefore, its catch was assumed nil for the Ecopath model.

We balanced the Ecopath model ensuring that: (i) EE values were less than unity, (ii) *P/Q* values were between 0.1 and 0.5 [39], (iii) production-to-respiration and respiration-to-assimilation ratios were less than unity, and (iv) the respiration-to-biomass ratios ranged between 1 and 10 for fish groups and higher for lower-trophic groups, in line with ecological and thermodynamic principles [40]. The pedigree index that classifies the model input data based on their sources was also calculated. The pedigree index scales between zero and one, and assigns high values to data from local sampling-based studies with high precision and low values to empirically estimated parameters and statistically collected data. A pedigree index close to one indicates higher input-data quality. Furthermore, we used pre-balance (PREBAL) diagnostics to assess the data quality of the input data in the Ecopath model [41]. PREBAL expects a linear positive slope for *B*, *P/B*, *Q/B* and *P/Q* from higher-trophic-level groups/species to lower-trophic-level groups/species.

The model results were evaluated using the ecosystem's statistical properties, calculated by capitalising on flows in the food web. Total system throughput (TST), which is the sum of all flows related to consumption, respiration, exports and detrital flows in the ecosystem, shows the size of the ecosystem and is akin to gross domestic product (GDP) in economic terms [42]. Furthermore, relative ascendancy and overhead (resilience) of the ecosystem were calculated. Ascendancy is a measure of the ecosystem's organisation and overhead is the strength of the ecosystem to resist stress [43]. Ascendancy is the power of an ecosystem to recover from perturbed conditions and resilience is the strength of its immune system; therefore, a balanced degree of ascendancy and resilience is required in an healthy ecosystem [44]. Furthermore, the ratios of total primary production to total respiration (Pp/R), total primary production to total biomass (Pp/B), total biomass to TST (B/T), and net system production, which is the difference between total primary production and system respiration, were calculated. In mature ecosystems, Pp/R is expected to approach unity, Pp/B is expected to be low, B/T, i.e., the amount of biomass supported per unit of energy, is expected to be high and net system production is expected to be close to zero [45].

Synthetic ecological indicators were also calculated using flows in the food web. Finn's cycling index (FCI) and Finn's mean path length (PL) were calculated. FCI shows the relative amount of flows being cycled in the food web, and PL is the average number of groups that a unit of flow (inflow or outflow) passes through, and both are expected to be high in mature ecosystems [46]. Furthermore, the predatory cycling index (PCI), which is

the proportion of TST cycled excluding the detritus compartment, was also calculated. The mean trophic level of the catch ($mTL_c$) and relative amount of primary production required to sustain fisheries' catches ($PPR_c$) were calculated to delineate the fisheries' impact on the ecosystem [47]. When an ecosystem is first fished, the $mTL_c$ is high and the $PPR_c$ is low and, as fishing intensifies, the $mTL_c$ is expected to decrease, creating a fishing down the food web effect [48], and the $PPR_c$ is expected to increase [49]. The system omnivory index (SOI) was also calculated to quantify the breadth of feeding interactions in the food web. The SOI is high when the ecosystem includes species/groups with a high variety of prey items in their diets, and low when the ecosystem is comprised of specialised consumers.

Transfer efficiencies between trophic levels in the food web were analysed using Lindeman spines, which show mean transfer efficiencies of energy flows between trophic levels by grouping flows and biomasses by integer trophic levels [50]. Mixed trophic impact (MTI) analysis was used to delineate the interactions between groups/species in the system. MTI analysis shows the direct and indirect trophic impacts between functional groups [51] and can be considered a prognostic analysis showing what would happen to other groups/species if a given group's/species' biomass in the system increases or decreases. A direct impact between groups/species occurs due to prey–predator interactions, e.g., the prey has a positive impact on its predator and the predator has a negative impact on its prey. An indirect impact occurs due to competition for the same resources or when a group/species has a direct impact on a prey or a predator of the other group, and this indirect impact outcompetes, if any, the direct impacts between the two groups. The value of MTI scales between $-1$, a strong negative impact, and 1, a strong positive impact. Furthermore, the keystoneness index (KS) was calculated to define keystone groups that have relatively low biomasses but structuring roles in the food web [52].

## 3. Results

### 3.1. Stomach Content Analysis

A total of thirteen stomachs were empty: seven in winter, three in spring, one in summer and two in autumn. The shortest specimen with an empty stomach was 11 cm, and the longest specimen with an empty stomach was 21 cm. Benthic crustaceans constituted the main prey of *N. randalli* and was dominated by shrimps, prawns and crabs. Considering fish species, the diet of *N. randalli* included species from the Clupeidae, Serranidae, Leiognathidae and Sparidae families (Table 2). The stomach contents of *N. randalli* intriguingly included *S. undosquamis*.

**Table 2.** Relative diet composition by weight of *N. randalli* in the Lamas region.

| Group | Diet Item | Weight (%) |
|---|---|---|
| Crustaceans | *Squilla* spp. | 26.99 |
| | *Charybdis longicollis* | 10.95 |
| | Unidentified crabs | 8.53 |
| | Stamatopoda | 6.11 |
| | *Penaeus japonicus* | 4.26 |
| | *Penaeus kerathurus* | 2.95 |
| | Unidentified shrimps | 1.28 |
| | Unidentified crustaceans | 1.2 |
| | *Macropthalmus* spp. | 0.94 |
| | *Penaeus* spp. | 0.53 |
| | Alpheidae | 0.12 |
| | Other Decapoda | 0.09 |

**Table 2.** *Cont.*

| Group | Diet Item | Weight (%) |
|---|---|---|
| Fish | *Clupea* spp. | 12.32 |
| | Unidentified teleost fish | 8.93 |
| | *Serranus hepatus* | 4.49 |
| | *Equulites elongatus* | 2.53 |
| | *Vanderhorstia mertensi* | 1.78 |
| | *Saurida undosquamis* | 1.44 |
| | Sparidae | 1.12 |
| Echinoderms | *Ophiaderma longicaudum* (Bruzelius, 1805) | 0.75 |
| | *Anseropoda placenta* | 0.11 |
| | Other Echinodermata | 0.02 |
| Other | Lophotrochozoa | 1.88 |
| | Digested organic material | 0.39 |
| | Endoparasites | 0.29 |

### 3.2. The Model

The Ecopath model of the Lamas region included 21 functional groups from phytoplankton with the lowest trophic level, to *M. merluccius* and Lessepsian *S. undosquamis* with the two highest trophic levels (Table 3). The majority of living biomass in the system was in TLs I, II and III (34.68%, 51.9% and 12.43%, respectively). The flow diagram of the Lamas region Ecopath model is given in Figure 2.

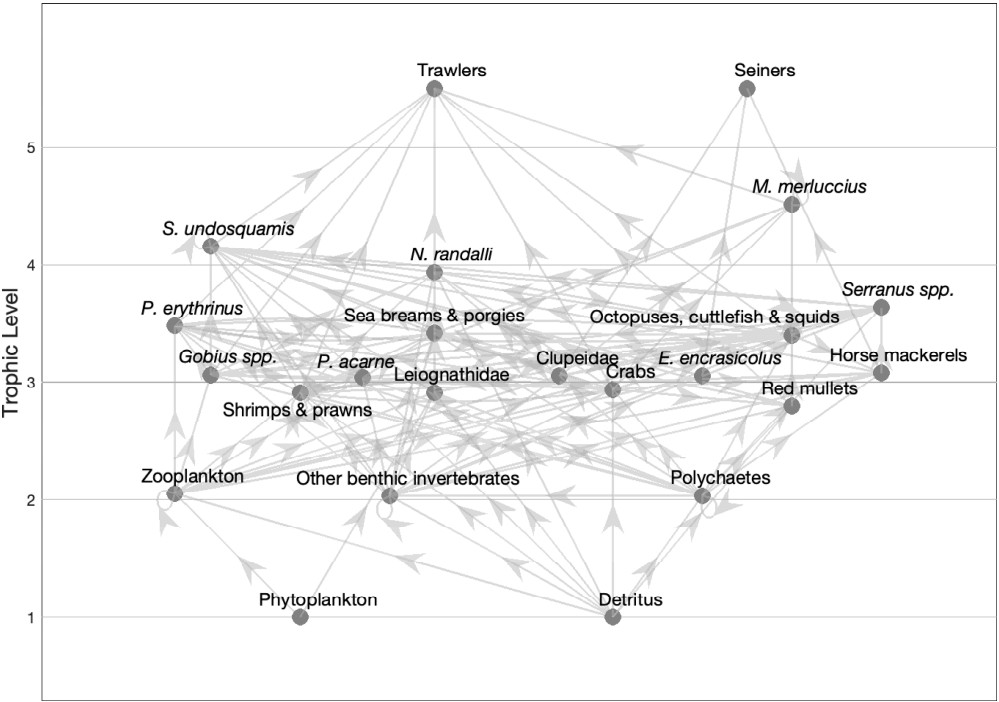

**Figure 2.** The flow diagram of the Lamas region Ecopath model.

**Table 3.** Input and output (bold) parameters of the Lamas region Ecopath Model.

| Group/Species | Trophic Level (TL) | Biomass (t km$^{-2}$) | P/B (y$^{-1}$) | Q/B (y$^{-1}$) | EE | P/Q | Landings (Tonnes km$^{-2}$ y$^{-1}$) By Seiners | By Trawlers |
|---|---|---|---|---|---|---|---|---|
| Phytoplankton | 1 | 7.75 | 195.1 | | **0.14** | | | |
| Zooplankton | 2.05 | 3.385 | 21.9 | 92.18 | **0.43** | **0.24** | | |
| *N. randalli* | 3.94 | 0.037 | 0.94 | 7.78 | **0.45** | **0.12** | | |
| Other benthic invertebrates | 2.03 | 5.456 | 1.15 | 3.66 | **0.87** | **0.31** | | |
| Polychaetes | 2.04 | 3.24 | 3.61 | 16.93 | **0.85** | **0.21** | | |
| Crabs | 2.94 | 0.618 | 2.42 | 5.53 | **0.93** | **0.44** | | $6.51 \times 10^{-5}$ |
| Shrimp and prawns | 2.91 | 0.251 | 3.09 | 11.27 | **0.95** | **0.27** | | 0.016 |
| Octopuses, cuttlefish and squids | 3.40 | 0.061 | 2.65 | 14.22 | **0.93** | **0.19** | | 0.008 |
| *P. erythrinus* | 3.48 | 0.084 | 1.77 | 8.43 | **0.88** | **0.21** | | 0.009 |
| *P. acarne* | 3.04 | 0.161 | 1.94 | 10.69 | **0.83** | **0.18** | | |
| Red mullets | 2.80 | 0.315 | 1.23 | 9.89 | **0.89** | **0.12** | | 0.013 |
| *M. merluccius* | 4.51 | 0.021 | 2.41 | 7.11 | **0.93** | **0.34** | | 0.001 |
| *Gobius* spp. | 3.06 | 0.36 | 1.69 | 11.07 | **0.85** | **0.15** | | 0.001 |
| *S. undosquamis* | 4.16 | 0.083 | 1.76 | 8.29 | **0.04** | **0.21** | | 0.001 |
| Sea breams and porgies | 3.41 | 0.293 | 0.41 | 7.65 | **0.91** | **0.05** | | 0.018 |
| *Serranus* spp. | 3.63 | 0.086 | 1.28 | 10.21 | **0.96** | **0.13** | | |
| Leiognathidae | 2.91 | 0.408 | 0.96 | 19.36 | **0.38** | **0.05** | | |
| Clupeidae | 3.05 | 0.447 | 1.28 | 14.21 | **0.86** | **0.09** | 0.042 | |
| *E. encrasicolus* | 3.05 | 0.07 | 2.73 | 12.23 | **0.84** | **0.22** | 0.001 | |
| Horse mackerels | 3.08 | 0.095 | 1.67 | 11.8 | **0.36** | **0.14** | 0.003 | |
| Detritus | 1 | 105.35 | | | **0.11** | | | |

### 3.3. Model Data Quality

The pedigree index of the Lamas region Ecopath model was 0.63, indicating a high level of input data quality.

The PREBAL analysis showed that the model input parameters conformed to a linear increasing trend from high to low trophic levels (Figure 3). The biomass values of sea breams and porgies, the octopuses, cuttlefish and squids group, *E. encrasicolus*, *P. acarne*, crabs, Leiognathidae, the shrimps and prawns group, and red mullets could be underestimated, whereas the biomass values of zooplankton, polychaetes, other benthic invertebrates and phytoplankton could be overestimated. The *P/B* values of sea breams and porgies, Clupeidae, Leiognathidae, red mullets, polychaetes and other benthic invertebrates could be underestimated, whereas the *P/B* values of *M. merluccius*, zooplankton and phytoplankton could be overestimated. The *Q/B* values of crabs and other benthic invertebrates could be underestimated, whereas the *Q/B* value of zooplankton could be overestimated. Finally, the *P/Q* values of *N. randalli*, *Serranus* spp., sea breams and porgies, Clupeidae, Leiognathidae and red mullets could be underestimated, whereas the *P/Q* values of *M. merluccius*, crabs and other benthic invertebrates could be overestimated.

### 3.4. Model Summary Statistics

The summary statistics of the Ecopath model are given in Table 4. The TST consisted of 12.6%, 37.8%, 7.2% and 42.4% of consumption, export, respiratory flows and flows into detritus compartments, respectively. The system's net primary production, net system production, and Pp/R and Pp/B ratios were high and the B/T ratio was low.

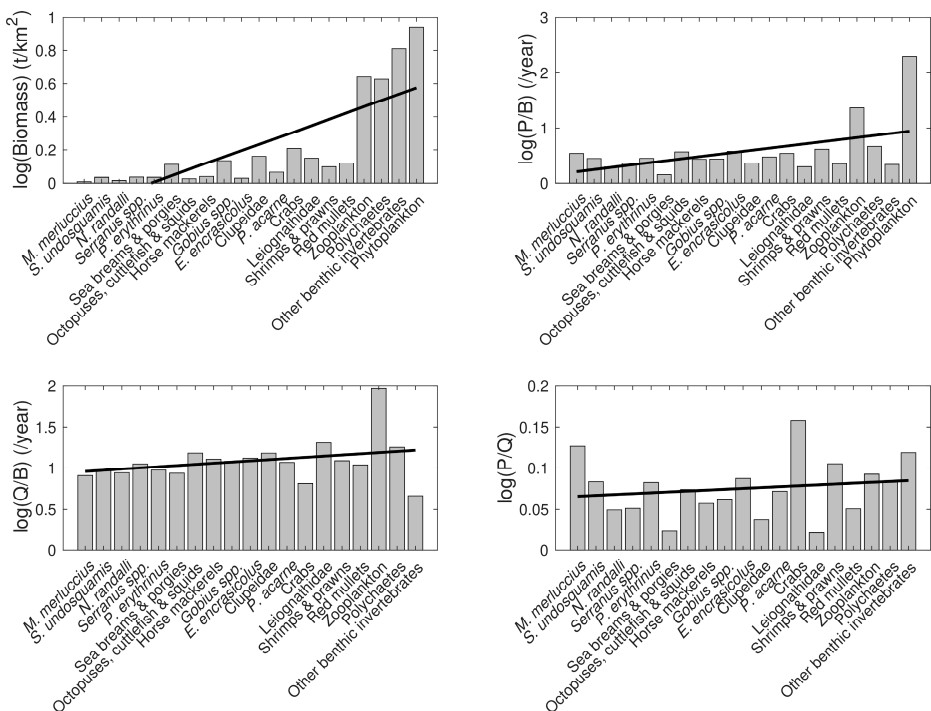

**Figure 3.** PREBAL analysis of the input parameters for the Lamas region Ecopath model.

**Table 4.** System summary statistics of the Lamas region Ecopath model.

| Parameter | Value | Unit |
|---|---|---|
| Sum of all consumption | 423.93 | t/km$^2$/year |
| Sum of all exports | 1270.62 | t/km$^2$/year |
| Sum of all respiratory flows | 241.38 | t/km$^2$/year |
| Sum of all flows to detritus | 1424.15 | t/km$^2$/year |
| Total system throughput | 3360.08 | t/km$^2$/year |
| Sum of all production | 1609.76 | t/km$^2$/year |
| Total net primary production | 1512 | t/km$^2$/year |
| Net system production | 1270.62 | t/km$^2$/year |
| Total biomass (excluding detritus) | 23.22 | t/km$^2$ |
| Total biomass/total throughput (B/T) | 0.007 | year |
| Total primary production/total respiration (Pp/R) | 6.26 | - |
| Total primary production/total biomass (Pp/B) | 65.11 | /year |
| Transfer efficiency from primary producers | 5.77 | % |
| Transfer efficiency from detritus | 12.56 | % |
| Mean transfer efficiency (TE) | 9.77 | % |
| Connectance index | 0.30 | - |
| System omnivory index (SOI) | 0.13 | - |
| Shannon diversity index | 1.90 | - |
| Total catch | 0.11 | t/km$^2$/year |
| Mean trophic level of catch (mTL$_c$) | 3.14 | - |
| Gross efficiency (catch/net primary production) | 0.0001 | - |
| Mean trophic level of community ($\geq$3.25) | 3.61 | - |
| Primary production required to sustain catches (PPR$_c$) | 0.68 | % |
| Predatory cycling index (PCI) | 3.49 | % |
| Finn's cycling index (FCI) | 2.24 | % |
| Finn's mean path length (PL) | 2.22 | - |
| Ascendancy | 48.3 | % |
| Overhead | 51.7 | % |
| Capacity | 7954 | flowbits |
| Ecopath pedigree index | 0.63 | - |

Considering the impact of fisheries on the ecosystem, the $mTL_c$ and mean trophic level of community (TL $\geq$ 3.25) were less than four, and the $PPR_c$ and the total fisheries catch were low.

Regarding the food web dynamics, the SOI was low, the connectance index was at moderate levels and the Shannon's diversity index was low. The transfer efficiency of energy from primary producers was lower than the transfer efficiency from detritus, and, overall, the transfer efficiency of energy along the food web was close to the theoretical ecological value of 10%. The PCI and FCI were at moderate levels, and the PL was low.

The capacity of the Lamas region marine ecosystem consisted of a balanced degree of system ascendancy and overhead.

*3.5. Mixed Trophic Impact Analysis*

The MTI analysis was performed to show interactions between groups in the food web (Figure 4). *N. randalli* had direct negative impacts on crabs, shrimps and prawns, *Serranus* spp., sea breams and porgies, and Leiognathidae and Clupeidae groups due to being their predator. It had indirect negative impacts on the indigenous *M. merluccius* due to competition. Although *N. randalli* is a prey to Lessepsian *S. undosquamis*, it had a negative impact on this species because *N. randalli*'s direct positive impact on *S. undosquamis* due to being its prey was outcompeted by its indirect negative impact due to competition for similar prey items. *N. randalli* had indirect positive impacts on indigenous *P. erythrinus*, *P. acarne* and red mullets due to negatively impacting important predators of these species, i.e., *S. undosquamis* and *M. merluccius*, because of competition. *N. randalli* had indirect positive impact on the octopuses, cuttlefish and squids group due to its indirect negative impact on predators of this group, i.e., sea breams and porgies, and *S. undosquamis*. *N. randalli* had indirect positive impacts on *E. encrasicolus* and horse mackerels due to its negative impacts on the predators of these groups, i.e., *S. undosquamis* and *Serranus* spp. Although *N. randalli* is a predator of *Gobius* spp., it had an indirect positive impact on this group due to its negative impacts on *Gobius* spp.'s predators, i.e., sea breams and porgies, *S. undosquamis* and *Serranus* spp. Finally, *N. randalli*'s direct negative impacts due to predation on other benthic invertebrates and polychaetes were outcompeted by its indirect positive impacts, i.e., the negative impacts of *N. randalli* on their main predators, namely, sea breams and porgies, *Serranus* spp., Leiognathidae and Clupeidae.

Lessepsian *S. undosquamis* is a predator of *N. randalli*, and therefore had a direct negative impact. *S. undosquamis* had strong direct negative impacts on indigenous fish species, i.e., *P. acarne*, sea breams and porgies, *Serranus* spp., *E. encrasicolus* and horse mackerels due to predation. *S. undosquamis* had an indirect negative impact on indigenous *M. merluccius* due to competition. *S. undosquamis* had positive impacts on *Gobius* spp., red mullets, octopuses, cuttlefish and squids, and shrimps and prawns, because its direct negative impacts, i.e., predation on these groups, were outcompeted by its indirect positive impacts due to negatively impacting competitors and/or other predators of these groups. *S. undosquamis* had an indirect positive impact on *P. erythrinus* because of its direct negative impact on one of the predators of this group, i.e., sea breams and porgies, due to being their predator.

The indigenous sea breams and porgies group had direct negative impacts on *P. erythrinus*, *P. acarne*, octopuses, cuttlefish and squids, and red mullets due to predation. The sea breams and porgies group had an indirect negative impact on Lessepsian *S. undosquamis*, due to competition, and an indirect positive impact on *Serranus* spp., due to their negative impact on *S. undosquamis*, which is a predator of *Serranus* spp. Furthermore, the sea breams and porgies group had a direct positive impact on *M. merluccius* as prey. The sea breams and porgies group had indirect positive impacts on *E. encrasicolus* and horse mackerels due to negatively impacting their main predator, i.e., *S. undosquamis*.

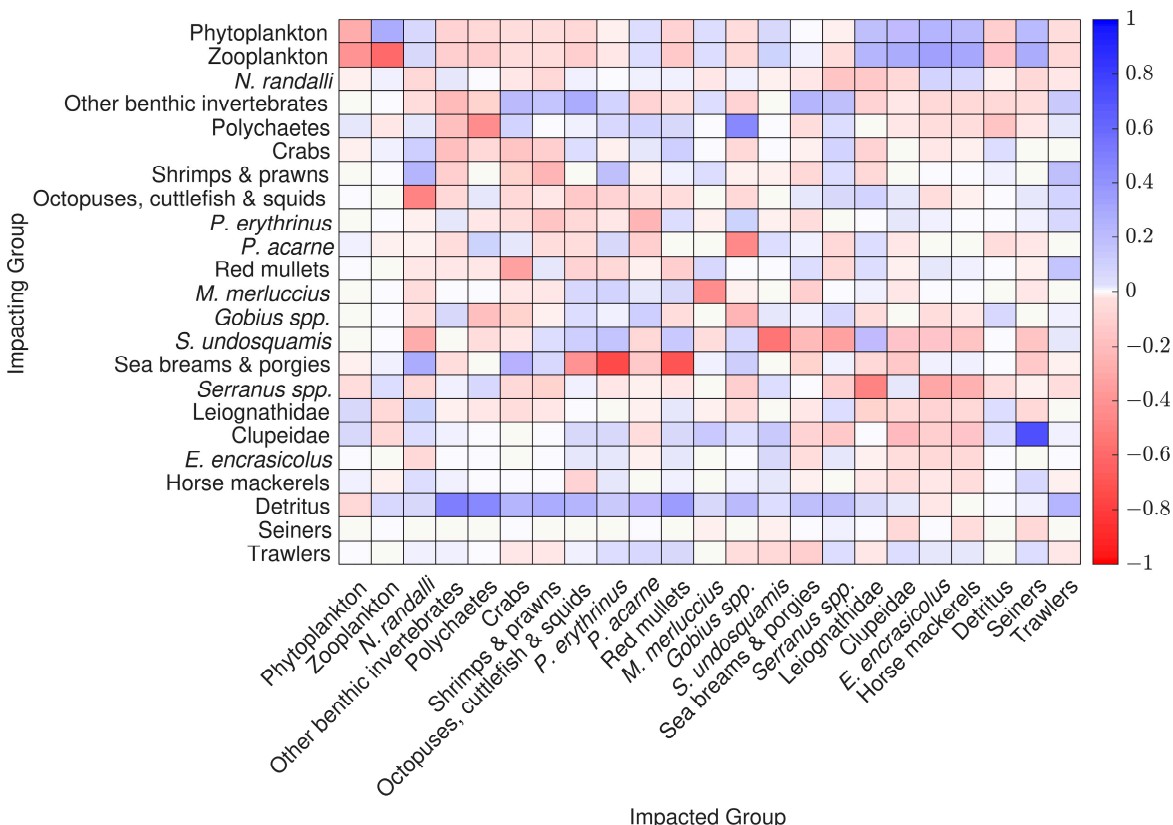

**Figure 4.** Mixed trophic impact analysis of the Lamas region Ecopath model.

Trawlers had direct negative impacts on crabs, shrimps and prawns, *Gobius* spp., *S. undosquamis*, sea breams and porgies, and *M. merluccius* due to exploitation, and an indirect negative impact on Leiognathidae due to exploiting their main predator, *S. undosquamis*. Trawlers had indirect positive impacts on *P. erythrinus*, *P. acarne*, red mullets, octopuses, cuttlefish and squids, and *Serranus* spp. because their direct negative impacts due to exploiting these species/groups were outcompeted by their indirect positive impacts, i.e., exploiting their predators, namely *M. merluccius* and *S. undosquamis*. Furthermore, trawlers had indirect positive impacts on Clupeidae, *E. encrasicolus* and horse mackerels due to exploiting their predators, i.e., *S. undosquamis* and *M. merluccius*. Seiners had direct negative impacts on Clupeidae and horse mackerels due to fisheries exploitation and indirect negative impacts on *S. undosquamis* and *M. merluccius* because of exploiting the main prey of these groups, i.e., Clupeidae, *E. encrasicolus* and horse mackerels. Although seiners directly exploited *E. encrasicolus*, they had an indirect positive impact because the indirect negative impact of seiners on *E. encrasicolus*'s predator, *S. undosquamis*, outcompeted their direct negative impact.

### 3.6. Keystoneness Analysis

The keystoneness index (KS) was used to identify functional groups and species that have a structuring role on the food web dynamics (Figure 5). The sea breams and porgies group had the highest keystone index value of 0.07 and the highest relative total impact (1.0) in the Lamas region ecosystem. *Serranus* spp. and zooplankton groups had the second highest keystone index value of −0.19, with relative total impact values of 0.55 and 0.64, affecting many of the groups/species as a predator and prey, respectively. Lessepsian *S. undosquamis* had the third highest keystone index value of −0.20, with a relative total impact of 0.54. Two pelagic species, namely *E. encrasicolus* and horse mackerels, had the lowest keystone index values of 0.95 and 0.96, and relative total impact values of 0.096 and 0.092, respectively.

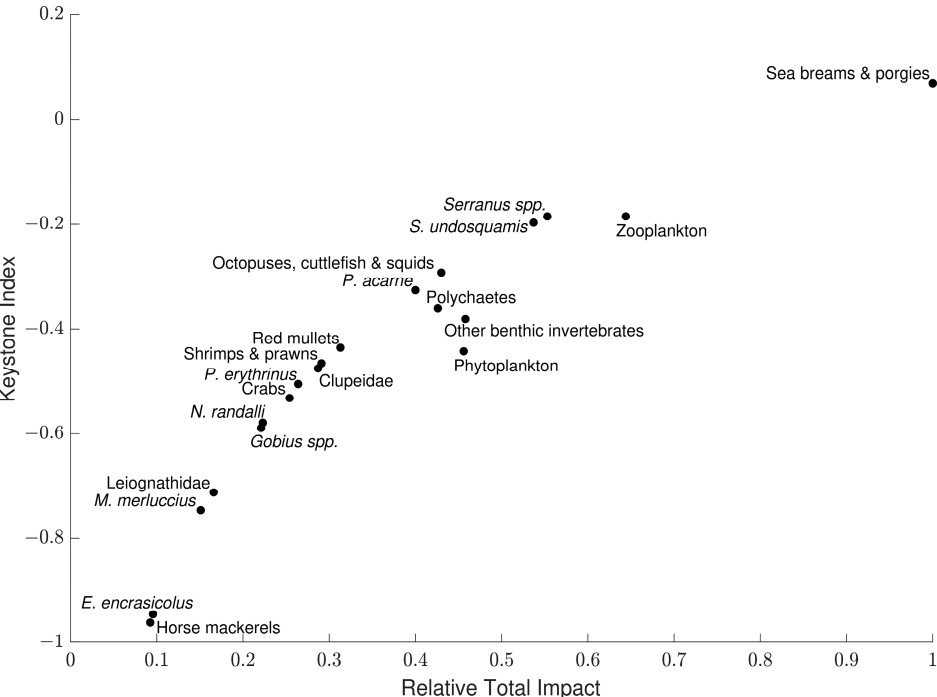

**Figure 5.** Keystoneness analysis of the species and functional groups in the Lamas region Ecopath model.

*3.7. Energy Flows*

The flows through trophic levels consisted of 41.3% of flows from detritus and 28.7% of flows from primary producers, indicating the dominance of the grazing food chain (Figure 6). The transfer efficiencies of flows were below 10% from TL II to TL III, above 10% from TL III to TL IV and close to 10% from TL IV to TL V. The highest respiratory flows and flows to detritus occurred from TL II. Exports were highest at TL III due to fisheries exploitation. The biomasses gradually decreased from TL II to TL V.

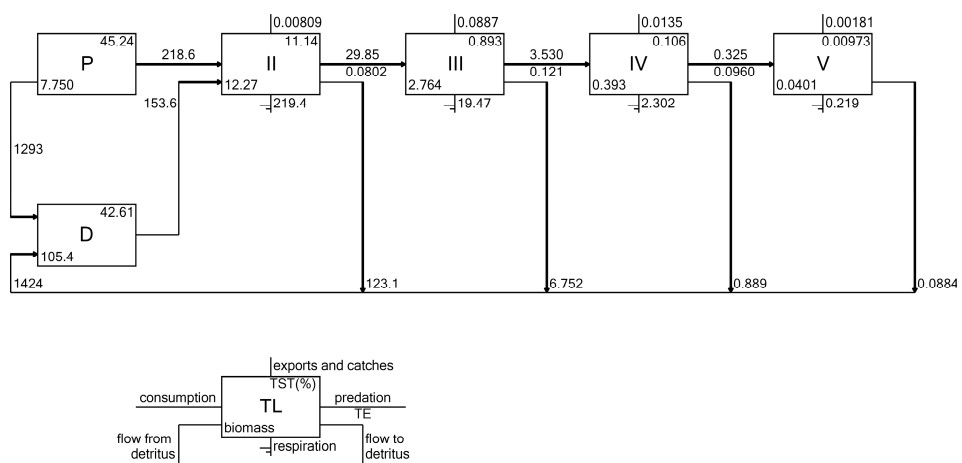

**Figure 6.** Lindeman spine graph showing energy flows (t/km$^2$/y) originating from primary producers and detritus compartments and biomasses (t/km$^2$) across trophic levels in the Lamas region Ecopath model.

## 4. Discussion

Our study highlighted that: (i) the diet composition of *N. randalli* mainly comprised invertebrates and fish, and the smooth brittle star was identified as a diet item for the first time; (ii) the ecosystem of the Lamas marine region was in a development state *sensu* [45]

and characterised by a high net system production and Pp/R ratio, and relatively low FCI and PL indicators; (iii) the keystone group in the ecosystem was the indigenous sea breams and porgies group, having a strong structuring role in the ecosystem; (iv) *N. randalli* had indirect positive impacts on commercially exploited native demersal fish species due to its mitigating role against the predation exerted by Lessepsian *S. undosquamis* on the demersal fish assemblages; and (v) *N. randalli* had a negative impact on the keystone group of the ecosystem, i.e., sea breams and porgies, and poses a risk to the ecosystem if this negative impact intensifies in the future.

Stomach content analysis showed that approximately 50% of the stomachs during the winter season were empty. Therefore, time of sampling is important for stomach content studies and indicated a difficulty for *N. randalli* in finding prey during winter. The diet items, namely *Charybdis longicollis*, *Squilla mantis*, *Penaeus* spp. and Echinodermata species, identified in this study were due to the overlap between *N. randalli*'s natural habitat and those species. Similar to earlier studies in the region [28,53], our study found that the subphylum Crustacea constituted the main prey items of *N. randalli*. However, the smooth brittle star (*Ophiaderma longicaudum*) was identified for the first time in *N. randalli*'s stomach in the eastern Mediterranean Sea. The presence of *S. undosquamis* in the stomach contents of *N. randalli* was hypothesised to be due to consumption in the trawl's cod-end during the haul rather than a natural phenomenon; therefore, the species was not included in the diet of *N. randalli* in the Ecopath model. Furthermore, the presence of endoparasites in the stomach contents was remarkable. Therefore, the impact of endoparasites on *N. randalli* and its possible effects on human health due to consumption should be analysed in future studies, as the species has attained increasing commercial importance in recent years [28].

The majority of the published Ecopath models had a pedigree index value between 0.4 and 0.59, and only 10% of the models had pedigree values that were above 0.6 [54]. The calculated pedigree index of the Lamas region Ecopath model was high and indicated a high degree of data quality for the Ecopath model because we capitalised on local sampling data for calculating the majority of the biomasses, and population and diet studies from the region, especially for fish groups. PREBAL analysis indicated some over- and underestimated parameters for certain groups, especially for the parameters that were borrowed from other models in the region or empirically calculated. In addition, the pedigree analysis of biomasses could have been affected by groups with high levels of aggregation at lower trophic levels, e.g., zooplankton and phytoplankton. Therefore, the respective increasing linear trend could be overestimated; however, because we balanced the models in line with thermodynamics and ecosystem theory [39,40], the Ecopath model of the Lamas region could be considered successful in representing the ecosystem conditions in the area.

A suite of summary statistics and synthetic ecological indicators of the Lamas region Ecopath model are given in Table 5 in comparison with other ecosystems in the Mediterranean Sea. The high net system production and Pp/R ratio indicated that the ecosystem of the Lamas region was in a developmental stage *sensu* [45]. The lower values of Pp/R calculated in previous modelling studies in other ecosystems across the Mediterranean Sea were due to the low levels of primary production modelled, although respiratory flows were similar. Furthermore, due to the higher levels of primary production estimated in our study, the TST was higher than other studies in the region; however, it was comparable to the values in the whole Mediterranean Sea, and North and Central Adriatic Sea ecosystems (Table 5). The relative ascendancy and overhead (resilience) values were balanced contrary to the other Mediterranean ecosystems, except the whole Mediterranean Sea ecosystem. In heathy ecosystems, a balanced degree of ascendancy and resilience is required to recover from perturbed conditions and to withstand against stress, respectively [44]. Therefore, the status of the Lamas region ecosystem could be considered healthy with respect to ascendancy and resilience indicators, although it is in a developmental stage based on net system production and high Pp/B and low B/T ratios, and the ecosystem experienced an autotrophic succession considering the Pp/R ratio.

Table 5. Summary statistics for the Lamas region Ecopath model in comparison with other regions in the Mediterranean Sea.

| Indicators | This Study | Israeli Coast [17] | Cyprus Coast [16] | North Aegean Sea [10] | Pagasitikos Gulf, Aegean Sea [13] | Thermaikos Gulf, Aegean Sea [12] | Saronikos Gulf, Aegean Sea [14] | North Aegean Sea [11] | North and Central Adriatic Sea [55] | South Catalan Sea [56] | Mersin Bay, Levant Sea [18] | Mediterranean Sea [39] | Unit |
|---|---|---|---|---|---|---|---|---|---|---|---|---|---|
| Year | 2019 | 2008–2012 | 2015.2017 | 2003–2006 | 2008 | 1998–2000 | 1998–2000 | 1993 | 1990s | 1994 | 2009–2013 | 2000s | |
| Sum of all respiratory flows | 241.38 | - | - | 269.48 | 486 | 417 | 571 | 271.68 | 421.09 | 327.16 | 254.63 | 290 | $t/km^2/year$ |
| Sum of all flows to detritus | 1424.15 | - | - | 562.53 | 761 | 868 | 1297 | 566.46 | 1387.46 | 416.91 | 292.12 | 1467 | $t/km^2/year$ |
| Total system throughput | 3360.08 | 631.89 | 841 | 1976 | 2951 | 3185 | 3925 | 1984.75 | 3844 | 1657 | 1149.53 | 4000 | $t/km^2/year$ |
| Total net primary production | 1512 | - | - | 535.48 | 712 | 923 | 1243 | 535.47 | 1149.85 | 386.68 | 368.65 | 1610 | $t/km^2/year$ |
| Net system production | 1270.62 | - | - | 265.99 | 227 | 506 | 672 | 263.80 | 728.76 | 59.52 | 114.2 | 1320 | $t/km^2/year$ |
| Total biomass (excluding detritus) | 23.22 | 8.69 | 18.77 | 33.04 | 78 | 40 | 38.94 | 33.98 | 130.3 | 59.99 | 23.49 | 42.74 | $t/km^2$ |
| Total biomass/total throughput | 0.007 | - | - | 0.02 | 0.03 | 0.01 | 0.01 | 0.02 | 0.03 | 0.04 | 0.02 | - | year |
| Total primary production/total respiration | 6.26 | 4.26 | 2.04 | 1.99 | 1.47 | 2.21 | 2.17 | 1.97 | 2.73 | 1.18 | 1.45 | 5.55 | - |
| Mean transfer efficiency | 9.77 | 19 | 16.93 | 17.4 | - | - | 14.77 | - | 10 | 12.6 | 9.37 | 9.2 | % |
| Connectance index | 0.30 | - | - | - | - | - | 0.332 | 0.28 | - | 0.20 | 0.27 | 0.1 | - |
| System omnivory index | 0.13 | 0.19 | 0.23 | 0.18 | 0.25 | 0.2 | 0.23 | 0.24 | 0.19 | 0.19 | 0.16 | 0.27 | - |
| Total catch | 0.11 | 0.93 | 0.65 | 2.35 | - | - | 2.75 | 2.93 | 2.44 | 5.36 | 0.42 | - | $t/km^2/year$ |

Table 5. *Cont.*

| Indicators | This Study | Israeli Coast [17] | Cyprus Coast [16] | North Aegean Sea [10] | Pagasitikos Gulf, Aegean Sea [13] | Thermaikos Gulf, Aegean Sea [12] | Saronikos Gulf, Aegean Sea [14] | North Aegean Sea [11] | North and Central Adriatic Sea [55] | South Catalan Sea [56] | Mersin Bay, Levant Sea [18] | Mediterranean Sea [39] | Unit |
|---|---|---|---|---|---|---|---|---|---|---|---|---|---|
| Mean trophic level of catch | 3.14 | 3.37 | 3.25 | 3.47 | - | - | 3.36 | 3.47 | 3.07 | 3.12 | 3.29 | 3.08 | - |
| Gross efficiency (catch/net primary production) | 0.0001 | - | - | 0.004 | - | - | 0.002 | 0.01 | 0.002 | 0.0014 | 0.001 | 0.00026 | - |
| Primary production required to sustain catches | 0.68 | 11.34 | 7.07 | 3.45 | - | - | 4.66 | - | 6.59 | 9.45 | 6.79 | 1.46 | % |
| Predatory cycling index | 3.49 | - | - | - | - | - | 14.77 | - | 3.97 | 3.33 | 3.67 | 10.96 | % |
| Finn's cycling index | 2.24 | 5.78 | 9.3 | 14.6 | - | - | 12.53 | - | 14.7 | 25.19 | 10.09 | 4.98 | % |
| Finn's mean path length | 2.22 | 2.63 | 3.21 | - | - | - | 3.121 | - | 5.41 | 4.27 | - | - | - |
| Ascendancy | 48.3 | - | - | 21.6 | - | - | 24.5 | - | 27 | 25.5 | 24.32 | 42.9 | % |
| Overhead | 51.7 | - | - | 78.4 | - | - | 75.5 | - | 73 | 74.5 | 75.66 | 57.1 | % |
| Capacity | 7954 | - | - | 9162.5 | - | - | 15,785 | - | 15,406.7 | 7119.3 | 4773.98 | - | flowbits |
| Ecopath pedigree index | 0.63 | 0.54 | 0.62 | 0.61 | 0.53 | 0.53 | 0.65 | - | 0.66 | 0.67 | 0.63 | - | - |

The FCI and PL were low, indicating significant flows to detritus from lower trophic levels. Indeed, a significant input to detrital compartments was calculated at TL II (Figure 6). The values of these two indicators increase as ecosystems develop, and high values are expected in mature ecosystems [45]. Therefore, the FCI and PL indicated the developmental status of the Lamas region ecosystem. The overall transfer efficiency of energy was close to the theoretical value, i.e., 10%, and lower from primary producers and higher from detritus compartments. Furthermore, the grazing food chain dominated the flows, again indicating the developmental status of the ecosystem.

Although the mTL$_c$ and PPR$_c$ were lower than those in the majority of the Mediterranean Sea ecosystems, mTL$_c$ was similar to those in the South Catalan Sea and whole Mediterranean Sea ecosystems. Furthermore, the total fisheries catch was orders of magnitude lower than those in the majority of the Mediterranean Sea ecosystems, indicating a low degree of fisheries impact.

Ref. [18] found that *N. randalli* had negative impacts on *P. erythrinus* and *P. acarne*. In addition, [17] reached a similar result, capitalizing on a modelling study with two aggregated functional groups defined as new alien demersal fishes and small indigenous demersal fishes, which included *N. randalli*, *P. erythrinus* and *P. acarne*, respectively. *N. randalli* had a positive and a negative impact on *S. undosquamis* in [17,18], respectively. Contrary to those previous findings, in our study, although *N. randalli* had similar dietary requirements and therefore, competed with *P. erythrinus*, *P. acarne* and, to some extent, red mullets, it had positive impacts on these species/groups due to its negative impact on the Lessepsian predator *S. undosquamis*. Therefore, MTI analysis suggested that *N. randalli* developed a favourable mitigating role in the food web against the negative impact of *S. undosquamis* on commercially important indigenous demersal species as a predator. However, the keystone group in the Lamas region ecosystem, i.e., sea breams and porgies, was negatively impacted by *N. randalli*; therefore, attention should be paid to the interaction between these two groups/species in future studies because an increase in the biomass of *N. randalli* in the region may instigate a reorganisation in the food web by obliterating the dynamics of sea breams and porgies.

*N. randalli* was suggested to have a high potential of being invasive [26]. Although *N. randalli* bears the potential to impact the ecosystem significantly, as shown in our MTI analysis, its impacts on other species in the food web have not yet reached the limits of causing biodiversity loss in the ecosystem. Therefore, *N. randalli* can be considered an alien species with the potential of being invasive, and ecosystem-based management activities should consider pre-emptive measures. On the Turkish coasts of the Mediterranean and Aegean Seas, *N. randalli* has increasingly been sold as *P. erythrinus* [28], and the species can be beneficial for the industrial fishery in the region. Furthermore, by capitalising on experiences employed against invasive aquatic species in other regions [19], fisheries can be utilised as a management tool to alleviate negative impacts exerted by *N. randalli* on the food web in addition to other management measures. However, a former modelling study on alien lionfish in the Mexican Caribbean showed that a suite of management measures is required to control populations of alien species successfully, such as restoration of habitat conditions to the advantage of indigenous species, regulating fisheries on native fish populations to increase competition and predation pressure on alien species [57]. If management strategies are not put in place, *N. randalli* may further establish its population with increasing biomass levels. With an optimistic outlook, *N. randalli* may cause a decrease in the native species' populations that it competes with or feeds on, causing an economical loss in fisheries, or, with a pessimistic outlook, it could trigger a cascading effect in the food web and cause a reorganisation in the ecosystem due to its negative impact on the ecosystem's keystone group, i.e., sea breams and porgies, hence creating unprecedented ecological and economic losses in the region.

In Gökova Bay in the Aegean Sea, the amount of Lessepsian fish in the landings was 22%, and its economic value constituted 9.6% of the economic value of the landings in 2019, and *N. randalli* constituted 12.8% of landings, and its economic value was 6.3% of

landings [58]. Therefore, *N. randalli* has already started to become an important commercial species in the catches and should be increasingly exploited to prevent its negative impacts on the indigenous species. However, the mitigating role of *N. randalli* in regulating the negative impacts of *S. undosquamis* on native small demersal fish species as well as *M. merluccius* could be adversely affected by its increasing exploitation, and future modelling studies should employ scenario simulations to assess the changes in this mitigating effect under different harvesting scenarios.

The Mediterranean is a transition region with a temperate climate influenced by a colder and wetter European climate and a warmer and drier African climate; therefore, it is a critical region for future climate changes [59]. Global warming is expected to increase seawater temperatures, and this creates a risk for native species to be replaced by Lessepsian species such as *N. randalli* in the Mediterranean Sea. Increasing sea temperatures in the Red and the Mediterranean Seas [60,61] in the recent decades have created more favourable conditions for the Red Sea species in the Mediterranean Sea [62,63] and facilitated an increase in the number of tropical species [64]. Therefore, the trophic impacts of, in general Lessepsian species and in particular *N. randalli*, on the food web of the northeastern Mediterranean Sea will likely increase as climate change can favour thermophilic species [11].

Fisheries management is complicated in the northeastern Mediterranean Sea [65]. However, several methods can be applied to mitigate the negative impacts of *N. randalli* in the eastern Mediterranean Sea. Targeted exploitation or a bounty system can be promoted to decrease the negative impact of *N. randalli* on the native species. Incentives for marketing of *N. randalli* can be another management strategy to decrease its population. Furthermore, implementation of marine protected areas (MPAs) can be used as a management strategy to decrease Lessepsian species' impacts with species-targeted removals. However, there is still debate about the impact of MPAs on invasive species, as MPAs in the northeastern Mediterranean Sea are already dominated (concerning number of species and biomass) by invasive species of Lessepsian origin [66,67]. Therefore, although *N. randalli* cannot be considered an invasive alien species yet, it has the potential to pose significant risks to the ecosystem, and its negative impacts should be counteracted by employing ecosystem-based fisheries management (EBFM) strategies.

*Limitations and Future Considerations*

The Lamas region Ecopath model capitalised on local sampling data from bottom trawl hauls. Bottom trawling is a fishing practice that can efficiently sample the demersal environment but cannot retain pelagic organisms effectively, and pelagic species either are underestimated or can be totally missing in the samples. Furthermore, benthic species are not effectively retained by bottom trawl nets. Therefore, in our Ecopath model of the Lamas region, certain groups could have been underestimated in the ecosystem. In addition, our model did not include any temporal dynamics due to lack of time series fish stock assessment studies that can be used to validate the temporal model; therefore, it was not possible to test fisheries management strategies under different harvesting levels and changing environmental conditions. Future modelling studies should work towards including temporal dynamics and scenario simulations to assess different fisheries management options for mitigating the negative impacts of *N. randalli*.

## 5. Conclusions

This study is the first species-specific Ecopath model showing the impact of one of the most common Lessepsian species, *N. randalli*, on the northeastern Mediterranean Sea food web using the Lamas region as a case study. In addition to assessing the ecosystem status and functioning by capitalising on synthetic ecological indicators and network analysis, we focused on the impact of *N. randalli* and its interaction with other species and fisheries. Our study used the most recent available data to describe the current state of the ecosystem functioning and structure in the region. We found that *N. randalli* had both the potential to mitigate negative impacts of other Lessepsian species in the region and to instigate

significant negative changes in the ecosystem. Targeted exploitation could be implemented to control the population of *N. randalli*; however, it may also bring adverse impacts, as *N. randalli* mitigates the negative effects of certain Lessepsian fish species on indigenous ones. Therefore, future modelling work should include temporal scenario simulations to delineate the impact of *N. randalli* on the ecosystem under different harvesting, food web and climate change scenarios.

**Author Contributions:** Conceptualization, E.A.; methodology, E.A. and Y.A.; formal analysis, Y.A. and E.A.; investigation, Y.A.; data curation, Y.A.; validation; E.A., writing—original draft preparation, Y.A. and E.A.; writing—review and editing, E.A. and Y.A.; visualization, E.A. and Y.A.; supervision, E.A. All authors have read and agreed to the published version of the manuscript.

**Funding:** This research was funded by The Scientific and Technological Research Council of Türkiye (TÜBİTAK), project number 117Y396.

**Institutional Review Board Statement:** The animal study protocol was approved by the Institutional Ethics Committee of Middle East Technical University (protocol code 2017/05 and date of approval 21 August 2017).

**Informed Consent Statement:** Not applicable.

**Data Availability Statement:** The data presented in this study are available in the tables provided in the article and in Appendix A.

**Acknowledgments:** The authors thank Meltem Ok for leading the field sampling and laboratory analyses, the crew of R/V Lamas 1 for facilitating the field operations and Gülşah Can for her guidance in laboratory analysis.

**Conflicts of Interest:** The authors declare no conflict of interest. The funders had no role in the design of the study; in the collection, analyses or interpretation of data; in the writing of the manuscript; or in the decision to publish the results.

## Appendix A

**Table A1.** Data and related sources regarding the input parameters for the Lamas region Ecopath model.

| Functional Groups | Original Value | Calibrated Value | Sources |
|---|---|---|---|
| **Phytoplankton** | | | |
| Biomass | 7.75 | 7.75 | [68] |
| *P/B* | 195.1 | 195.1 | Calculated to match 151.2 gC/m$^2$/y annual primary production as per [69] |
| **Zooplankton** | | | |
| Biomass | 3.385 | 3.385 | [70] |
| *P/B* | 30.42 | 30.42 | [71] |
| *Q/B* | 92.18 | 92.18 | [18] |
| Diet | [18] | | |
| ***N. randalli*** | | | |
| Biomass | 0.0374 | 0.0374 | Trawl survey |
| *P/B* | 0.936 | 0.936 | [72] |
| *Q/B* | 9.781 | 7.781 | Empirical equation by [35] using length–weight relationship and $L_\infty$ values from [73] and aspect ratio from trawl survey |
| Diet | Stomach content analysis, [28,53,74] | | |
| **Other benthic invertebrates** | | | |
| Biomass | 0.0546 | 5.456 | Trawl survey |
| *P/B* | 1.15 | 1.15 | [10] |
| *Q/B* | 3.658 | 3.658 | [10] adjusted with Opitz's correction factor [75] |
| Diet | Modified from [10] | | |

| Functional Groups | Original Value | Calibrated Value | Sources |
|---|---|---|---|
| **Polychaetes** | | | |
| Biomass | 1.62 | 3.24 | [76] |
| *P/B* | 3.61 | 3.61 | [18] |
| *Q/B* | 16.93 | 16.93 | [10] adjusted with Opitz's correction factor [75] |
| Diet | Modified from [10] | | |
| **Crabs** | | | |
| Biomass | 0.0618 | 0.618 | Trawl survey |
| *P/B* | 2.42 | 2.42 | [10] |
| *Q/B* | 5.526 | 5.526 | [10] adjusted with Opitz's correction factor [75] |
| Diet | Modified from [10] | | |
| **Shrimps and prawns** | | | |
| Biomass | 0.251 | 0.251 | Trawl survey |
| *P/B* | 3.09 | 3.09 | [18] |
| *Q/B* | 12.27 | 11.27 | [18] |
| Diet | Modified from [10] | | |
| **Octopuses, cuttlefish and squids** | | | |
| Biomass | 0.102 | 0.0612 | Trawl survey |
| *P/B* | 2.652 | 2.652 | [10] |
| *Q/B* | 14.22 | 14.22 | [10] adjusted with Opitz's correction factor [75] |
| Diet | [77] | | |
| *P. erythrinus* | | | |
| Biomass | 0.140 | 0.0837 | Trawl survey |
| *P/B* | 1.769 | 1.769 | [78] |
| *Q/B* | 8.432 | 8.432 | Empirical equation by [35] using length–weight relationship and $L_\infty$ (as (weighted average of max length divided by 0.95 [79]) values from [80] |
| Diet | [81] | | |
| *P. acarne* | | | |
| Biomass | 0.230 | 0.161 | Trawl survey |
| *P/B* | 1.94 | 1.94 | [82] |
| *Q/B* | 10.69 | 10.69 | Empirical equation by [35] using length–weight relationship and $L_\infty$ values from [83] |
| Diet | [83] | | |
| **Red mullets** | | | |
| Biomass | 0.45 | 0.315 | Trawl survey |
| *P/B* | 1.225 | 1.225 | [78] |
| *Q/B* | 9.894 | 9.894 | Empirical equation by [35] using length–weight relationship and $L_\infty$ values from [78] |
| Diet | [84] | | |
| *M. merluccius* | | | |
| Biomass | 0.0209 | 0.0209 | Trawl survey |
| *P/B* | 2.41 | 2.41 | [85] |
| *Q/B* | 7.115 | 7.115 | Empirical equation by [35] using length–weight relationship and $L_\infty$ values from [85] |
| Diet | [86] | | |
| *Gobius* **spp.** | | | |
| Biomass | 0.0018 | 0.360 | Trawl survey |
| *P/B* | 0.847 | 1.695 | Empirical equation by [34] using maximum age value from [87]. |
| *Q/B* | 11.07 | 11.07 | Empirical equation by [35] using length–weight relationship and $L_\infty$ values from [87] |
| Diet | [87] | | |
| *S. undosquamis* | | | |
| Biomass | 0.0829 | 0.0829 | Trawl survey |
| *P/B* | 1.76 | 1.76 | [88] |
| *Q/B* | 8.285 | 8.285 | Empirical equation by [35] using length–weight relationship from [89] and $L_\infty$ from [88] |
| Diet | [90] | | |

**Table A1.** *Cont.*

| Functional Groups | Original Value | Calibrated Value | Sources |
|---|---|---|---|
| **Sea breams and porgies** | | | |
| Biomass | 0.419 | 0.293 | Trawl survey |
| *P/B* | 0.415 | 0.415 | Empirical equation by [34] capitalising on weighted averages of calculated Z values using maximum age values from [36] |
| *Q/B* | 8.654 | 7.654 | Empirical equation by [35] using length–weight relationship from [91] and $L_\infty$ from [92] |
| Diet | [93] | | |
| ***Serranus* spp.** | | | |
| Biomass | 0.0123 | 0.0864 | Trawl survey |
| *P/B* | 1.28 | 1.28 | [94] |
| *Q/B* | 12.215 | 10.215 | Empirical equation by [35] using $W_\infty$ from [95] |
| Diet | [96] | | |
| **Leiognathidae** | | | |
| Biomass | 0.408 | 0.408 | Trawl survey |
| *P/B* | 0.961 | 0.961 | [97] |
| *Q/B* | 19.36 | 19.36 | Empirical equation by [35] using $W_\infty$ from [97] |
| Diet | [98] | | |
| **Clupeidae** | | | |
| Biomass | 0.00224 | 0.447 | Trawl survey |
| *P/B* | 1.282 | 1.282 | [99] |
| *Q/B* | 14.21 | 14.21 | Empirical equation by [35] using length–weight relationship and $L_\infty$ from [99] |
| Diet | [100] | | |
| ***E. encrasicolus*** | | | |
| Biomass | 0.00703 | 0.0703 | Trawl survey |
| *P/B* | 2.73 | 2.73 | [18] |
| *Q/B* | 12.23 | 12.23 | Empirical equation by [35] using length–weight relationship and $L_\infty$ from [101] |
| Diet | [102] | | |
| **Horse mackerels** | | | |
| Biomass | 0.095 | 0.095 | Trawl survey |
| *P/B* | 1.66 | 1.6 | [101] |
| *Q/B* | 11.80 | 11.80 | [101] |
| Diet | [39,40] | | |
| **Detritus** | | | |
| Biomass | 105.4 | 105.4 | Empirical equation by [103] using 151.2 gC/m$^2$/y primary production and euphotic zone depth of 37 from [69] |

**Table A2.** Relative diet composition matrix for the Lamas region Ecopath model.

| # | Prey/Predator | 2 | 3 | 4 | 5 | 6 | 7 | 8 | 9 | 10 | 11 | 12 | 13 | 14 | 15 | 16 | 17 | 18 | 19 | 20 |
|---|---|---|---|---|---|---|---|---|---|---|---|---|---|---|---|---|---|---|---|---|
| 1 | Phytoplankton | 0.7 | | | | | | | | | | | | | | | 0.03 | | | |
| 2 | Zooplankton | 0.05 | | | | 0.02 | 0.11 | | 0.1 | 0.338 | | | 0.11 | 0.01 | 0.284 | 0.15 | 0.725 | 0.97 | 1.0 | 0.975 |
| 3 | *N. randalli* | | | | | | | 0.01 | | | | | | 0.01 | | | | | | |
| 4 | Other benthic invertebrates | | 0.009 | 0.005 | | 0.41 | 0.4 | 0.54 | 0.331 | 0.086 | 0.209 | | | 0.001 | 0.307 | 0.183 | 0.042 | 0.02 | | 0.003 |
| 5 | Polychaetes | | 0.003 | 0.027 | 0.034 | 0.42 | 0.22 | 0.09 | 0.107 | 0.106 | 0.235 | | 0.863 | | 0.037 | 0.029 | 0.102 | 0.01 | | 0.001 |
| 6 | Crabs | | 0.227 | | | 0.02 | 0.025 | 0.175 | 0.076 | 0.039 | 0.174 | 0.001 | 0.027 | | 0.04 | 0.2 | | | | 0.003 |
| 7 | Shrimps and prawns | | 0.329 | | | 0.01 | 0.055 | 0.075 | 0.226 | 0.016 | | 0.114 | | 0.005 | 0.02 | 0.134 | | | | 0.003 |
| 8 | Octopuses, cuttlefish and squids | | | | | | | 0.02 | 0.023 | | | | | 0.001 | 0.041 | | | | | 0.015 |
| 9 | *P. erythrinus* | | | | | | | 0.01 | | | | | | | 0.05 | | | | | |
| 10 | *P. acarne* | | | | | | | 0.01 | 0.1 | | | | | 0.1 | 0.05 | | | | | |
| 11 | Red mullets | | | | | | | 0.005 | | | | 0.134 | | 0.035 | 0.126 | | | | | |
| 12 | *M. merluccius* | | | | | | | | | | | 0.31 | | | | | | | | |
| 13 | *Gobius* spp. | | 0.025 | | | | | 0.055 | 0.024 | 0.184 | | | | 0.028 | 0.009 | 0.104 | | | | |
| 14 | *S. undosquamis* | | | | | | | | | | | | | 0.008 | | | | | | |
| 15 | Sea breams and porgies | | 0.016 | | | | | | 0.013 | | | 0.164 | | 0.08 | | | | | | |
| 16 | *Serranus* spp. | | 0.062 | | | | | | | | | | | 0.128 | | | | | | |
| 17 | Leiognathidae | | 0.135 | | | | | | | | | | | 0.031 | | 0.1 | | | | |
| 18 | Clupeidae | | 0.17 | | | | | 0.01 | | | | 0.277 | | 0.378 | 0.036 | 0.01 | | | | |
| 19 | *E. encrasicolus* | | | | | | | | | | | | | 0.143 | | 0.07 | | | | |
| 20 | Horse mackerels | | | | | | | | | | | | | 0.051 | | 0.02 | | | | |
| 21 | Detritus Import | 0.25 | 0.026 | 0.968 | 0.966 | 0.12 | 0.19 | | | 0.232 | 0.382 | | | | | | 0.101 | | | |

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
