# Peer review of "Randall’s Threadfin Bream (Nemipterus randalli, Russell 1986) Poses a Potential Threat to the Northeastern Mediterranean Sea Food Web"

_fishes, doi:10.3390/fishes8080402_

Round 1

Reviewer 1 Report

The manuscript focuses on the community-wide effects of a single invasive species. The modelling tool is a standard approach and the paper is an excellent exercise for linking modeling to understanding ecology, with the hope of future applications. The paper is clearly written, also its English is great.

In the title, I would remove "Lessepsian" or I would say "The Lessepsian invasive..."

line 11-12: additional food may sound like a positive effect on fish but on the whole community it can have mixed effects, both positive and negative, to a range of other species. In a food web context mixed effects are guaranateed, with positive effects on some and negative effects on other species. Both can include fish species if you consider also indirect effects. Your approach is exacly targeting these issues, so in the Sbtract I would not over-emphasize the positive effect of giving food to fish.

line 77-79: can you give some examples here for invasive species and also a few for mitigations by fisheries? Just in a short paragraph.

Testing and validating the outcome of such models is not easy. One way is the comparative perspective. Is there any example for "similar" fish invading marine food webs in "similar" trophic positions? There are so many examples for EwE models and predictive simulations, you may try to find some examples. For example:

https://akjournals.com/view/journals/168/20/2/article-p161.xml

https://link.springer.com/article/10.1007/s42974-021-00069-0

Also, other but similar models could be helpful to consider (e.g. loop analysis):

https://journals.plos.org/plosone/article?id=10.1371/journal.pone.0130261

You may consider mentioning these examples and enriching a comparative perspective.

line 128-148: you describe nicely data sampling based on the food of the bream but the whole EwE model was constructed and parameterized based only on these information?

The functional groups are more or less taxonomic groups. Functional groups can be constructed in a more functional sense, not necessarily following taxonomy. Like instead of Polychaeta, you may have groups like "benthic macroinvertebrates", just for an example.

The EwE-based analysis is very nicely done and presented.

line 302: empty stomachs raise the question whether eDNA analysis could be helpful to complete our knowledge (I mean metabarcoding the stomach environment).

In the PREBAL analysis, the trend comes from the huge biomass of the strongly aggregated groups on the right. Without these huge aggregates, the high-resolution (low-aggregation) groups show no tendency, I think. You may mention that this result is highly sensitive to the aggregation level (and algorithm).

Table 4: these parameters really make sense only in a comparative context, as you show it in Table 5. Please make this richer, referring to other systems both statistically (like for the pedigree index) and biologically (mentioning the trophic level of other invasive species, for example).

Figure 5: again, aggregation: without the big aggregates, the relationship is almost linear, so the two indices are quite redundant. I would redraw this graph only for the functional groups including fish, forgetting the 4 big aggregates.

All in all, this is a very nice study.

Reviewer 2 Report

This MS discusses the biological invasions in marine ecosystems, which is interesting. Some modifications are needed as follows.

General comments:

1.The term “Lessepsian Randall’s threadfin bream” appears many times in the abstract and other sections in this MS. If the author is referring to a fish species, the Latin name is written in parentheses so that English + Latin name is used for the first time in the MS and Latin name is used in subsequent texts to reduce name repetition.

2.The abstract needs to be rewritten. The authors came up with some good findings using the EWE model, but they are not reflected in the abstract. The abstract would be better written according to the background and motivation for the study, the methods used, the main results and findings. Much of the present form is descriptive discourse.

3.There is a difference between biological invasions and alien species, and the former is certainly detrimental to local ecosystems. But for now, the authors point out that Lessepsian Randall's threadfin bream has both advantages and disadvantages for the local ecosystem, so does it count as a biological invasion? Do stakeholders need to prevent, control, and remove it? The authors need to clarify this point in the MS; the title is ambiguous.

Specific comments:

4.Materials and methods section: Separate subheadings for study area, data, and methods are required.

5.Line 286-287: Limitations of the discussion on trophic level, can it be linked to the mean trophic level (MTL) comparison or discussion? See https://doi.org/10.1038/nature09528. 

6.Table 3: Trophic level for each functional group/species need to be given in Table 3. Moreover, Seiners and Trawlers, Which one is the Landing (add unit)?

7.Line 605-621: If it is defined as an invasive species that is harmful to other fish stocks, then prevention and control is necessary from the perspective of ecological and sustainable management of fisheries, even if it is beneficial to some fish stocks.

8.Line 613: Marine Protected Areas (MPAs).

9.The final section of the conclusion can add the limitations of the model and the future research directions.

10.There is a large literature on EWE models, and it is recommended to update the recent literature.
